# CRACK IN THE ARMOR: UNIVERSAL STABILITY MEASUREMENT FOR LARGE LANGUAGE MODELS

## ABSTRACT

Large Language Models (LLMs) and Vision Language Models (VLMs) have become essential to general artificial intelligence, demonstrating impressive capabilities in task understanding and problem-solving. The real-world functionality of these large models critically depends on their stability. However, there is still a lack of rigorous studies examining the stability of LLMs when subjected to various perturbations. In this paper, we aim to address this gap by proposing a novel influence measure for LLMs. This measure is inspired by statistical methods grounded in information geometry, offering desirable invariance properties. Using this framework, we analyze the sensitivity of LLMs in response to parameter or input perturbations. To evaluate the effectiveness of our approach, we conduct extensive experiments on models of varying sizes, from 1.5B to 13B parameters. The results clearly demonstrate the efficacy of our measure in identifying salient parameters and pinpointing vulnerable areas of input images that dominate model outcomes. Our research not only enhances the understanding of LLM sensitivity but also highlights the broad potential of our influence measure in optimizing models for tasks such as model quantization and model merging.

## 1 INTRODUCTION

Large Language Models (LLMs) and Vision Language Models (VLMs) such as GPT (Brown et al., 2020) and Llama (Touvron et al., 2023), have revolutionized the field of Natural Language Processing (NLP), exhibiting remarkable proficiency across a variety of tasks (Achiam et al., 2023; Jiang et al., 2023) and modalities (Bai et al., 2023; Liu et al., 2024). These modern LLMs are massive in size, trained on vast amounts of data, and meticulously aligned to prevent from generating harmful content (Perez et al., 2022), leaking private information (Zhang et al., 2024), or exhibiting sexual or religious bias (Xie & Lukasiewicz, 2023).

Despite the enthusiasm for these integrative approaches, a critical issue remains: LLMs remain susceptible to both external and internal perturbations, affecting their reliability and performance. This paper seeks to address this challenge by systematically identifying and locating these vulnerabilities through *quantifying the stability of LLMs under perturbations in both input and parameters.*

Externally, LLMs are susceptible to input perturbations, such as carefully crafted prompts, making them vulnerable to jailbreak attacks (Chao et al., 2023). Previous studies (Yuan et al., 2023; Huang et al., 2023) have highlighted that specific prompts, suffixes, and even so-called "magic code" (Liu et al., 2023) can induce inappropriate or harmful behaviors in language models. This susceptibility also extends to visual inputs in VLMs, as demonstrated by Qi et al. (2024), where adversarially optimized images can drastically shift model behavior.

In addition to intentionally crafted adversarial samples, VLMs are also highly sensitive to undetectable perturbations in specific local regions of an image — an issue that is common, as user-uploaded images often suffer from blurring, masking, or low resolution. Our case study on the Qwen-VL model (Bai et al., 2023) illustrates this vulnerability. We applied our proposed influence measure to each pixel in the input image, as shown in the right columns of Figure 1. Using these calculations, we masked the top 10 most sensitive pixels that were irrelevant to the prompt question. This manipulation easily misled the model, resulting in incorrect responses. Even when we crafted prompts specifically instructing the model to ignore noise and occlusions (see Figure 2), vulnerable regions remained, leading the model to hallucinate.

Internally, the stability of LLMs is further challenged by parameter perturbations, often introduced through quantization and model merging. While these methods are essential for deployment efficiency by reducing inference costs (Frantar & Alistarh, 2023; Ashkboos et al., 2024) and training costs (Yu et al., 2024), they can lead to hallucinations and degraded performance (Men et al., 2024; Yu et al., 2023; Li et al., 2024). Existing approaches (Ma et al., 2023; Frantar et al., 2022) primarily rely on the Hessian matrix of parameters, which results in substantial computational overhead and introduces unrealistic assumptions, which hinders fine-grained analysis of parameter stability.

To address these challenges, we propose a novel influence measure called **FI**, **F**irst order local **I**nfluence, to quantitatively assess the stability of LLMs against local perturbations. This influence measure is highly versatile and capable of capturing model stability when exposed to both external and internal perturbations at various levels of granularity—from channels of parameter matrices and individual parameters to input features like pixels and patches.

Specifically, we construct a perturbation manifold that encompasses all perturbed models, along with its associated geometric properties. Our influence measure quantifies the degree of local influence of the perturbation to a given objective function within this manifold and thus reflects the stability of each component of LLMs. We then provide a detailed discussion of the theoretical and computational advantages brought by the invariance property of the FI measure, which is entirely free from the constraints imposed by perturbations.

Empirically, we find our approach successfully uncovers sensitivity issues in VLMs by detecting fragile input pixels and enables us to understand how prompt engineering affects model stability. Subsequently, we conduct extensive empirical studies to demonstrate the validity of our method in identifying salient parameters across models of varying sizes. By safeguarding the salient parameters identified by our measure, we achieve a significant reduction in degradation caused by model quantization and model merging, highlighting the crucial role of FI in model optimization tasks.

Our main contributions are summarized as follows:

(i) We construct a novel framework for sensitivity analysis that evaluates the stability of each component within LLMs, considering the impact of small perturbations on the overall system.

(ii) Using our proposed metrics, we empirically demonstrate how external perturbations impact model performance, presenting an example involving vision-language models that highlights issues of erroneous sensitivity. Additionally, we illustrate how our approach can be employed to analyze cross-modal influence.

(iii) We empirically demonstrate that the proposed method effectively identifies the sensitive components of model parameters that influence performance under perturbations. It can serve as a valuable tool for enhancing model quantization and merging techniques, providing both theoretical insights and empirical evidence.

## 2 LOCAL INFLUENCE MEASURE OF LARGE LANGUAGE MODELS

In this section, we propose a new metric called FI to quantify the stability of large language models against local perturbations. Specifically, we focus on auto-regressive language models, which generate one token at a time based on both the initial input and previously generated tokens. Considering this sequential generation nature, we first develop FI for single-step token generation tasks and discuss its theoretical and computational properties in great detail. Following this, we demonstrate how the proposed stability measure can be naturally extended to address sequence generation tasks.

**Problem formulation.** Consider an LLM parameterized by $\theta$, with input data $x$, which may consist of text or, for visual language models, a combination of text and images. Given $x$, the model generates a probability distribution over its vocabulary to predict the next token, which can be framed as a classification problem with $K$ classes, where $K$ represents the vocabulary size.

However, vocabulary sizes are typically large (Bai et al., 2023; Dubey et al., 2024), and predictions are often concentrated on a small subset of tokens. Instead of using the entire vocabulary, it is more efficient to focus on a relevant subset based on the task. For example, in multiple-choice questions, probabilities are restricted to the choices "A", "B", "C", or "D". Classes can also be defined

semantically, such as categorizing tokens as "neutral" or "notorious" in toxicity detection (Gehman et al., 2020).

With appropriately defined classes, the predicted probability for class $y \in \{1, \ldots, K\}$ is denoted as $P(y|x, \theta)$, satisfying $\sum_{y=1}^{K} P(y|x, \theta) = 1$. Let $\omega \in \mathbb{R}^d$ be a perturbation vector varies in an open subset $\Omega$. $\omega$ can be applied to a subset of the model parameters $\theta$ and locations within the input data $x$. We denote the output of the perturbed model under this perturbation as $P(y|x, \theta, \omega)$.

**FI metric.** Since our primary interest lies in examining the behavior of $P(y|x, \omega, \theta)$ as a function of $\omega$ near $\omega_0 = 0$, we shift focus from $\theta$ to $\omega$. We introduce the perturbation manifold as defined in Zhu et al. (2007) and Zhu et al. (2011).

**Definition 2.1.** *Define the d-dimensional perturbation manifold $\mathcal{M} = \{P(y|x, \theta, \omega) : \omega \in \Omega\}$, which encompasses all perturbed models. Assume that for all $\omega \in \Omega$, the perturbed models $\{P(y = i|x, \theta, \omega)\}_{i=1}^{K}$ are positive and sufficiently smooth. The tangent space $T_\omega$ of $\mathcal{M}$ at $\omega$ is spanned by the partial derivatives of the log-likelihood function $\ell(\omega|y, x, \theta) = \log P(y|x, \theta, \omega)$ with respect to $\omega$, specifically $T_\omega = span\{\frac{\partial}{\partial \omega_i} \ell(\omega|y, x, \theta)\}_{i=1}^{d}$.*

The metric $g_\omega$ on $\mathcal{M}$ can be defined with the metric tensor $G_\omega$. Let $v_j(\omega) = h_j^\top \partial_\omega \ell(\omega|y, x, \theta) \in T_\omega$ for $j = 1, 2$ be two tangent vectors at $\omega$, their inner product is defined as $\langle v_1(\omega), v_2(\omega) \rangle_{g_\omega} = \sum_{y=1}^{K} v_1(\omega) v_2(\omega) P(y|x, \theta) = h_1^\top G_\omega h_2$ with $G_\omega = \sum_{y=1}^{K} \partial_\omega \ell(\omega|y, x, \theta) \partial_\omega^\top \ell(\omega|y, x, \theta) P(y|x, \theta, \omega)$. Subsequently, the norm of $v_j(\omega)$ under metric $g_\omega$ is $\|v_j\|_{g_\omega} = \sqrt{h_j^\top G_\omega h_j}$. Let $C(t) = P(y|x, \theta, \omega(t))$ be a smooth curve on the manifold $\mathcal{M}$ connecting two points $\omega_1 = \omega(t_1)$ and $\omega_2 = \omega(t_2)$, then the distance between $\omega_1$ and $\omega_2$ along the curve $C(t)$ is given by

$$S_C(\omega_1, \omega_2) = \int_{t_1}^{t_2} \sqrt{\|\partial_t log(P(y|x, \theta, \omega(t)))\|_{g_\omega}} dt = \int_{t_1}^{t_2} \sqrt{\frac{d\omega(t)^T}{dt} G_{\omega(t)} \frac{d\omega(t)}{dt}} dt. \quad (1)$$

With the Perturbation manifold $\mathcal{M}$ and respective metric $g_\omega$ defined, we are ready to propose the metric that quantifies the stability of large language models (LLMs) against various types of local perturbations. Let $f(\omega)$ be the objective function of interest for sensitivity analysis, in our case being $-\log P(y_{pred}|x, \theta, \omega)$, we can define the following (first-order) local influence metric FI:

**Definition 2.2.** *Given the perturbation manifold $\mathcal{M}$ and its metric, the first-order local influence measure of $f(\omega)$ at $\omega(0) = \omega_0$ is defined as*

$$\mathbf{FI}_\omega(\omega_0) = \max_C \lim_{t \to 0} \frac{[f(\omega(t)) - f(\omega(0))]^2}{S_C^2(\omega(t), \omega(0))}. \quad (2)$$

The ratio in Equation 2 measures the amount of change introduced to the objective function relative to the distance of the perturbation on the perturbation manifold. Thus, Equation 2 can be naturally interpreted as the maximum local ratio of change among all possible perturbation curves $C(t)$. The proposed FI measure has the property of transformation invariance.

**Theorem 2.3** (Reparametrization invariance). *Suppose that $\phi$ is a diffeomorphism of $\omega$. Then, $FI_\omega(\omega_0)$ is invariant with respect to any reparameterization corresponding to $\phi$. Specifically, let $\tilde{\omega}(t) = \phi \circ \omega(t)$ and $\tilde{\omega}_0 = \phi(\omega_0)$, we have $FI_{\tilde{\omega}}(\tilde{\omega}_0) = FI_\omega(\omega_0)$ holds, where detailed proof can be found in Shu & Zhu (2019).*

Theorem 2.3 establishes that the $FI_\omega(\omega_0)$ is invariant under any diffeomorphic (e.g., scaling and spinning) reparameterization of the original perturbation. This invariance property is not shared by other measures, such as Jacobian norm (Novak et al., 2018), Cook's local influence measure (Cook, 1986) and Sharpness (Novak et al., 2018). For instance, consider a perturbation of the form $\alpha + \Delta\alpha$, where $\alpha$ is a subvector of $(x^\top, \theta^\top)^\top$. If we apply a scaling reparameterization $\alpha' = K \odot \alpha$, where $K$ is a scaling vector and $\odot$ denotes element-wise multiplication, then the Jacobian norms changes:

$$\|J(\alpha)\|_F = \left[ \sum_i \left( \frac{\partial f}{\partial \alpha_i} \right)^2 \right]^{1/2} \neq \|J(\alpha')\|_F. \text{ In contrast, the FI measure remains unchanged.}$$

This transformation invariance is crucial for establishing a meaningful stability measure for neural networks. Components of LLMs often exhibit symmetric configurations, leading to a continuum of parameter settings that yield identical model behavior (Dinh et al., 2017; Amari, 1998). Such symmetries commonly appear in structures like ReLU-activated multilayer perceptrons and attention modules. As noted in Dinh et al. (2017) and Neyshabur et al. (2015), the rectified linear activation function possesses the *non-negative homogeneity* property. Specifically, let $\phi_{\text{ReLU}}(m) = \max(m, 0)$, we have

$$\forall (m, k) \in \mathbb{R} \times \mathbb{R}^+, \quad \phi_{\text{ReLU}}(km) = k\phi_{\text{ReLU}}(m).$$

Consider parameters $\theta_1$ and $\theta_2$ of two consecutive MLP layers within the LLM $P(y|x, \theta)$. Focusing on the expression $\phi_{\text{ReLU}}(m \cdot \theta_1) \cdot \theta_2$ and ignoring other modules for simplicity, we define a family of transformations $T_k : (\theta_1, \theta_2) \to (\theta_1', \theta_2') = (k\theta_1, k^{-1}\theta_2)$ for any positive scalar $k$. Due to the non-negative homogeneity of the ReLU function, the network's behavior remains unchanged under these transformations:

$$\phi_{\text{ReLU}}(m \cdot \theta_1') \cdot \theta_2' = \phi_{\text{ReLU}}(m \cdot k\theta_1) \cdot (k^{-1}\theta_2) = \phi_{\text{ReLU}}(m \cdot \theta_1) \cdot \theta_2.$$

This invariance leads to non-identifiability of the parameters, as different parameter values produce the same mapping. However, stability measures like the Jacobian norm vary with different $k$ values, rendering them unsuitable. Indeed, the Jacobian norm approaches infinity as $k$ increases, which is problematic, especially when parameters have heterogeneous scales (see Shu & Zhu (2019) for further discussion). In contrast, the FI measure avoids this scaling issue by employing the metric tensor of the perturbation manifold instead of the standard Euclidean metric.

**Computation of FI.** As we will show, Theorem 2.3 on diffeomorphic reparameterization invariance enables us to derive an easy-to-compute solution for Equation 2, while addressing the low-dimensionality problem inherent in LLMs.

**Theorem 2.4.** *If $G_\omega$ is positive definite, the **FI** measure have the following closed form*

$$\mathbf{FI}_\omega(\omega_0) = \nabla_{f(\omega_0)}^T G_{\omega_0}^{-1} \nabla_{f(\omega_0)}, \tag{3}$$

*where* $\nabla_{f(\omega_0)} = \partial f(\omega) / \left. \partial \omega \right|_{\omega = \omega_0}$

The detailed proof of Theorem 2.4 can be found in Appendix A.1. It is important to note that the closed form of FI in Theorem 2.4 depends on the positive definiteness of $G_\omega$, which is not always guaranteed. This is due to the fact that the parameters in LLMs are often high-dimensional tensors with low-rank structures (Kaushal et al., 2023). Motivated by Shu & Zhu (2019), we apply the invariant Theorem 2.3 by transforming $\omega$ to a vector $\nu$ such that $G_\nu = \mathbf{I}_K$, where $K$ is an integer. Specifically, we notice that $G_{\omega_0} = B_0^T B_0$, where $B_0 = [P(y = i|x, \theta, \omega)^{1/2} \partial_\omega \ell(\omega|y = i, x, \theta)]_{1 \leqslant i \leqslant K}$. Let $r_0 = \text{rank}(G_{\omega_0})$, we apply the compact SVD to $B_0 \in \mathcal{R}^{p \times K}$, which yield $B_0 = V_0 \Lambda_0 U_0$, where $V_0 \in \mathbb{R}^{p \times r_0}$ and $U_0 \in \mathbb{R}^{r_0 \times K}$ are semi-orthogonal matrices and $\Lambda_0 \in \mathbb{R}^{r_0 \times r_0}$ is a diagonal matrix. Under the transformation $\nu = \Lambda_0 V_0^T \omega$, we have

$$\mathbf{FI}_\omega(\omega_0) = \mathbf{FI}_\nu(\nu_0) = \nabla_{f(\nu_0)}^T \nabla_{f(\nu_0)} = \nabla_{f(\omega_0)}^T (V_0 R_0)^T \Lambda_0^{-2} (V_0 R_0) \nabla_{f(\omega_0)}, \tag{4}$$

where the second equality holds by applying the chain rule to $G_\nu$.

**Average FI.** Since the perturbation manifold $\mathcal{M}$ depends on the input data $x$, FI is inherently a data-dependent measure. To make this dependence explicit, we denote $\mathbf{FI}_\omega(\omega_0)$ as $\mathbf{FI}(x, \theta, \omega)$. In analyses focusing on individual cases, such as pixel vulnerability analysis, where $\omega$ represents a subset of the input data $x$, a data-dependent stability measure suffices. However, when the focus shifts to assessing the stability of the model parameters alone, a data-independent measure is required.

To address this, we follow the approach of other stability metrics like the Jacobian norm (Novak et al., 2018) and define the average FI as $\mathbb{E}_{P_x}[\mathbf{FI}(x, \theta, \omega)]$, where $P_x$ represents the distribution of the input data $x$. Given a pre-collected dataset $\mathcal{D} = \{x_1, \ldots, x_n\}$ with $x_i \sim P_x$, the average FI can be approximated by the empirical FI: $\mathbb{E}_{\widehat{P}_x}[\mathbf{FI}(x, \theta, \omega)] = n^{-1} \sum_{i=1}^n \mathbf{FI}(x_i, \theta, \omega)$, where $\widehat{P}_x$ denotes the empirical distribution of $x$ based on the dataset $\mathcal{D}$.

**FI for sequence generation.** Sequence generation is essentially multiple rounds of next-token generation, where the $l$-th token $y^{(l)}$ is generated given the initial input $z$ and previously generated tokens $\{y^{(1)}, \ldots, y^{(l-1)}\}$. We define the FI measure for generating the $l$-th token $y^{(l)}$ given the initial input $z$ by averaging out the randomness from the preceding steps

$$\mathbf{FI}_l(z) = \mathbb{E}_{y^{(1)}, \ldots, y^{(l-1)}}\left[\mathbf{FI}(\{z, y^{(1)}, \ldots, y^{(l-1)}\}, \theta, \omega)\Big| z\right]. \tag{5}$$

To formulate an overall measure for sequence generation, we aggregate these per-token FI measures. Since sequences generated by LLMs can vary in length, we propose two methods to handle this heterogeneity. The first approach sets a fixed horizon $L$ and computes the mean FI over these rounds

$$\mathbf{FI}_{\text{seq}}^{L}(z) = \frac{1}{L} \sum_{l=1}^{L} \mathbf{FI}_l(z). \tag{6}$$

Alternatively, inspired by the concept of average discounted rewards in reinforcement learning (Liu et al., 2018), we consider sequences of potentially infinite length and propose a discounted FI measure with discount factor $\gamma$

$$\mathbf{FI}_{\text{seq}}^{\infty,\gamma}(z) = (1 - \gamma) \sum_{l=0}^{\infty} \gamma^l \cdot \mathbf{FI}_l(z).$$

By taking the expectation over the distribution of $z$, we obtain the average FI for sequence generation in both cases $\mathbb{E}_{P_z}\left[\mathbf{FI}_{\text{seq}}^{L}(z)\right]$ and $\mathbb{E}_{P_z}\left[\mathbf{FI}_{\text{seq}}^{\infty,\gamma}(z)\right]$, respectively.

## 3 EXPERIMENT

In this section, we first conduct an experiment using the FI to evaluate the model's stability to external perturbations, such as disturbing the input images of VLMs, while highlighting its application in cross-modal analysis. Next, we validate the proposed approach's effectiveness in performing sensitivity analysis in the presence of internal perturbations. Finally, we present a series of experiments to illustrate the potential applications of FIs in several critical problems related to LLMs, including model quantization and merging.

### 3.1 EXTERNAL PERTURBATIONS ANALYSIS

We first show how to compute the FI value for each pixel of an image input of VLM. This illustration highlights that pixels with high FI values are more susceptible to external perturbations. Specifically, the experiment is conducted using a multiple-choice problem sampled from the ScienceQA dataset (Lu et al., 2022) with the Qwen-VL model.

Let $X$ be an image input to a VLM with RGB channels. Given the input $X$ the VLM outputs the prediction probability for each of the choice $y \in \{1, \ldots, 4\}$ denoted by $P(y|x,\theta)$ and pick $y_{pred} = \arg\max_y P(y|x,\theta)$ as its answer. Let $\omega_i \in \mathbb{R}^3$ denote the perturbation vector to the $i$-th pixel, we focus the influence of that perturbation to the entropy of the prediction, defined as $f(x) = -\log P(y_{pred}|x,\theta)$. In Figure 1, we calculated the FI values for each pixel using Equation 4. The FI intensity of each pixel is displayed in the image on the far right. Notably, some pixels exhibit significantly higher FI values compared to others.

As shown in the right columns of Figure 1, the overall distribution of FI values across the pixels is non-uniform. The FI values in the background regions are relatively low, ranging between 0 and 1, while the FI values around relevant objects, such as the kelp and starfish, are comparatively higher, with certain regions reaching up to 4.9. However, not all areas covering these relevant objects are equally vulnerable. This phenomenon suggests that the model exhibits heightened sensitivity to perturbations in specific patches, rather than uniformly across the entire object.

To further investigate, we perturb the image by masking the top-10 FI value patches, as shown in the middle column of Figure 1. This perturbation does not cause the model to lose the essential information regarding the food chain relationships necessary to answer the question, but it still induces hallucinations. The model shifts from confidently providing the correct answer—bat star ($p = 0.72$) and kelp ($p = 0.28$)—to a less confident and incorrect response—bat star ($p = 0.43$) and kelp ($p = 0.57$). For comparison, we include four types of random noise in the appendix A.2, each leading to similar levels of information loss; however, they fail to induce the same error. This empirical finding demonstrates the usefulness of FI in detecting vulnerable inputs for VLMs, suggesting that more robust training methods can be developed by focusing on these sensitive regions.

**Cross-Modal Analysis.** To gain deeper insight into how prompt engineering affects model stability, we conducted experiments to explore the impact of different prompt engineering strategies.

We defined two distinct types of prompt settings:

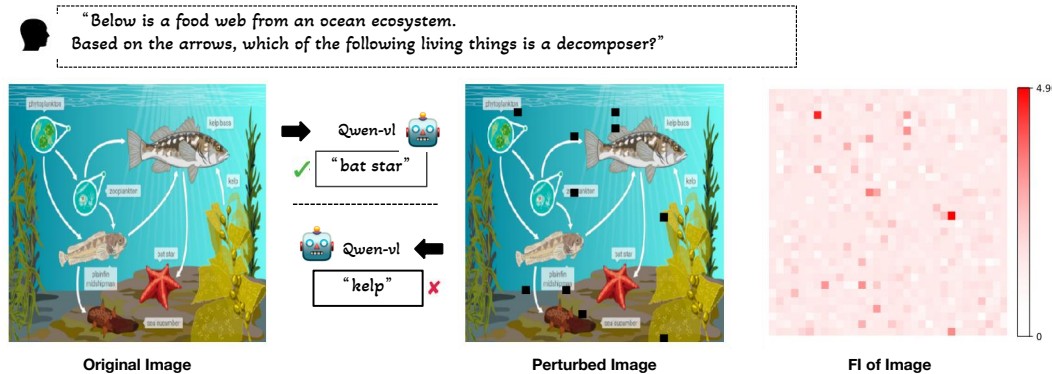

Figure 1: A case study of the Qwen-VL model on the SCI-QA dataset. The image on the far right shows the per-pixel FI values calculated for this case, while the middle image illustrates how perturbing the top 10 positions with the highest FI values induces hallucinations in the model.

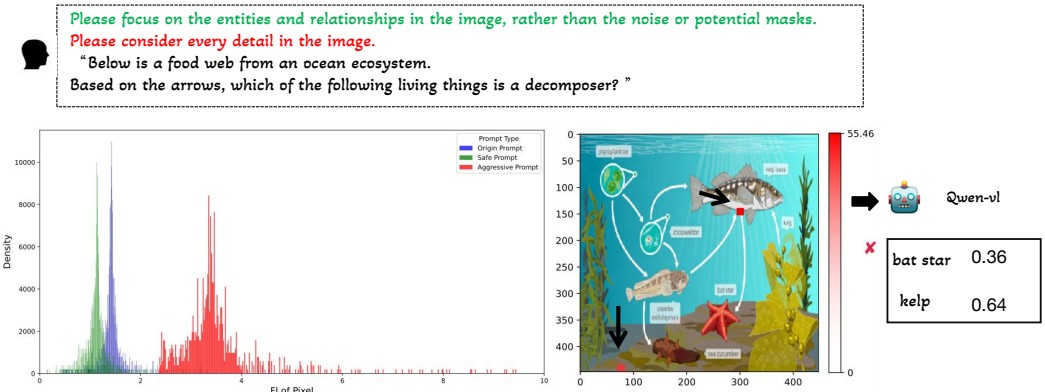

Figure 2: A case study utilizing FI for cross-modal analysis. In the same example, the bottom-left image shows how introducing control information with prompts like 'aggressive' and 'safe' affects the FI distribution on the image. The more cautious the prompt, the smaller the FI values.

- The "aggressive" prompt, such as *"Please consider every detail in the image,"* aims to enhance the model's focus on finer details.

- The "safe" prompt, such as *"Please focus on the entities and relationships in the image, rather than the noise or potential masks,"* reduces the model's attention to irrelevant noise.

We computed the Feature Importance (FI) value for each pixel and visualized how these values are distributed under different prompt settings. As illustrated in the left column of Figure 2, the FI values clearly reflect the prompt instructions. The "aggressive" prompt shifts the FI distribution to higher values, with a noticeable increase in both the mean and maximum FI values, indicating that the model becomes more sensitive to pixel-level perturbations across the image. Conversely, the "safe" prompt significantly shifts the FI distribution toward lower values, indicating reduced sensitivity and enhanced stability against perturbations in less relevant areas.

However, neither prompt fully guarantees model stability. As depicted in the right column of Figure 2, overlaying the FI values obtained under the "safe" prompt onto the image reveals that while the model adheres to the prompt by reducing its sensitivity to the background, it remains vulnerable to perturbations in two specific patches. Masking out these two patches still leads to incorrect answers. This experiment highlights the value of our framework in conducting cross-modal analysis and improving model robustness.

## 3.2 INTERNAL PERTURBATIONS ANALYSIS

Regarding the internal perturbations analysis, we begin by conducting a parameter sensitivity experiment. In this process, we compute FI values for each element in the weight matrices across all layers. We then perform sparsification by selecting a proportion of parameters with the highest FI values and setting them to 0-bit (i.e., zeroing them out).

For comparison, we randomly select the same proportion of parameters with lower FI values for sparsification. This approach helps us assess whether FI effectively identifies the most vulnerable parameters. Our analysis focuses on how these perturbations impact two key capabilities of large models: knowledge retention and instruction-following.

**Knowledge Retention Ability.** We conduct experiments on the multiple-choice problems from the MMLU (Hendrycks et al., 2020) dataset, using Qwen2-7B. We take the cross-entropy loss, *i.e.*, $f = -\log P(y = y_{\text{pred}}|x, \theta)$, as the target function, and calculate the FI value according to Theorem 2.4. In this setup, we treat the task as a 4-class classification problem with the possible classes being "A," "B," "C," and "D".

As shown in Figure 3, sparsifying just 2-3% of the high-FI parameters significantly reduces the model's knowledge capacity, inducing catastrophic forgetting and hallucinations, leading to up to a 75% performance loss. This demonstrates the effectiveness of FI values in identifying fragile parameters. In contrast, models remain relatively robust against random sparsification, where, in many cases, models exhibit almost the same behavior after a 5% sparsification. As this trend of performance decline manifests, a very small proportion of parameters dominate knowledge processing and retention.

**Instruction-following Ability.** We use the Alpaca-eval validation set (Dubois et al., 2024), a widely adopted benchmark, and conduct experiments with various open-source models, including LLaMA2, LLaMA3 (Touvron et al., 2023), and Qwen2 (Bai et al., 2023), across different sizes. We report two metrics: ROUGE-1 (comparing to pre-sparsity responses) and length-control winning rate (LCWR), comparing to GPT-3.5 Turbo. Higher scores are better for both metrics.

To estimate the average Fisher Information (FI) for sequence generation, we use the fixed-context approach with $L = 5$. For each sample $z$, we estimate $\mathbf{FI}_l(z)$ by generating $N = 10$ responses, truncating them at position $l - 1$. These truncated sequences are used to approximate the conditional expectation in Equation 5 by computing the sample average. The per-token FI values are then aggregated using Equation 6 to obtain $\mathbf{FI}_{\text{seq}}^L(z)$, which is averaged across all samples to estimate the overall FI.

Table 1 presents the results, clearly showing that models subjected to random sparsity consistently outperform those subjected to FI-guided sparsity. Notably, our FI metric demonstrates the greatest advantage over random selection at the 10% sparsity level. This is due to the skewed distribution of FI values, where a small subset of parameters exhibits significantly higher FI values, while the majority have comparatively low values (a visual analysis of this is provided in the appendix). This finding supports the existence of inherent structure within the parameter matrix, a phenomenon previously observed in the literature, though primarily in the context of rank analysis. Additionally, LLaMA models show lower sensitivity compared to Qwen models, likely due to the inclusion of supervised fine-tuning data during the pre-training phase of the LLaMA models. "

## 3.3 POTENTIAL APPLICATIONS OF FI IN LLM PROBLEMS

As we have demonstrated in the previous sections, FI is a powerful tool locating vulnerabilities within LLMs. In this section, we would take a step further, trying to safeguard these vulnerable points when model is exposed to perturbation. Specifically, we design and evaluate FI-guided safeguard methods for model quantization and merging.

**Application on Model Quantization.** Model quantization becomes a pivotal technique for the large-scale deployment of LLMs, as it significantly reduces memory consumption by lowering the precision of model parameters. However, this efficiency does not come without cost—there is an inherent trade-off between parameter precision and model performance. Existing quantization methods predominantly aim to mitigate the resulting loss in accuracy by employing increasingly sophisticated computational strategies (Frantar et al., 2022). In this work, we present a simple yet

Table 1: Performance of Different Models Based on Criteria with Full Value and Sparsity Percentages

| Model | Criteria | Full | 6% Sparsity | | 8% Sparsity | | 10% Sparsity | | 12% Sparsity | |
|---|---|---|---|---|---|---|---|---|---|---|
| | | | FI-High | FI-Low | FI-High | FI-Low | FI-High | FI-Low | FI-High | FI-Low |
| Llama2-13B | Rouge-1 | 1.0 | 0.52 | $0.59 \pm 0.02$ | 0.4 | $0.43 \pm 0.06$ | 0.18 | $0.68 \pm 0.01$ | 0.05 | $0.19 \pm 0.03$ |
| | LCWR | 0.43 | 0.38 | $0.41 \pm 0.03$ | 0.29 | $0.34 \pm 0.07$ | 0.09 | $0.42 \pm 0.0$ | 0.01 | $0.08 \pm 0.05$ |
| Llama3-8B | Rouge-1 | 1.0 | 0.46 | $0.52 \pm 0.04$ | 0.21 | $0.41 \pm 0.06$ | 0.09 | $0.25 \pm 0.04$ | 0.04 | $0.12 \pm 0.03$ |
| | LCWR | 0.42 | 0.4 | $0.38 \pm 0.01$ | 0.12 | $0.30 \pm 0.03$ | 0.0 | $0.12 \pm 0.01$ | 0.0 | $0.01 \pm 0.01$ |
| Llama2-7B | Rouge-1 | 1.0 | 0.44 | $0.56 \pm 0.01$ | 0.25 | $0.45 \pm 0.02$ | 0.06 | $0.33 \pm 0.02$ | 0.0 | $0.21 \pm 0.02$ |
| | LCWR | 0.42 | 0.32 | $0.4 \pm 0.0$ | 0.12 | $0.35 \pm 0.01$ | 0.0 | $0.19 \pm 0.05$ | 0.0 | $0.1 \pm 0.03$ |
| Qwen2-7B | Rouge-1 | 1.0 | 0.09 | $0.41 \pm 0.05$ | 0.01 | $0.30 \pm 0.09$ | 0.01 | $0.31 \pm 0.06$ | 0.01 | $0.15 \pm 0.02$ |
| | LCWR | 0.41 | 0.03 | $0.35 \pm 0.03$ | 0.02 | $0.25 \pm 0.1$ | 0.03 | $0.20 \pm 0.05$ | 0.03 | $0.08 \pm 0.02$ |
| Qwen2-1.5B | Rouge-1 | 1.0 | 0.18 | $0.4 \pm 0.13$ | 0.16 | $0.32 \pm 0.02$ | 0.05 | $0.28 \pm 0.08$ | 0.05 | $0.23 \pm 0.02$ |
| | LCWR | 0.14 | 0.03 | $0.07 \pm 0.04$ | 0.04 | $0.02 \pm 0.02$ | 0.0 | $0.04 \pm 0.0$ | 0.0 | $0.02 \pm 0.02$ |

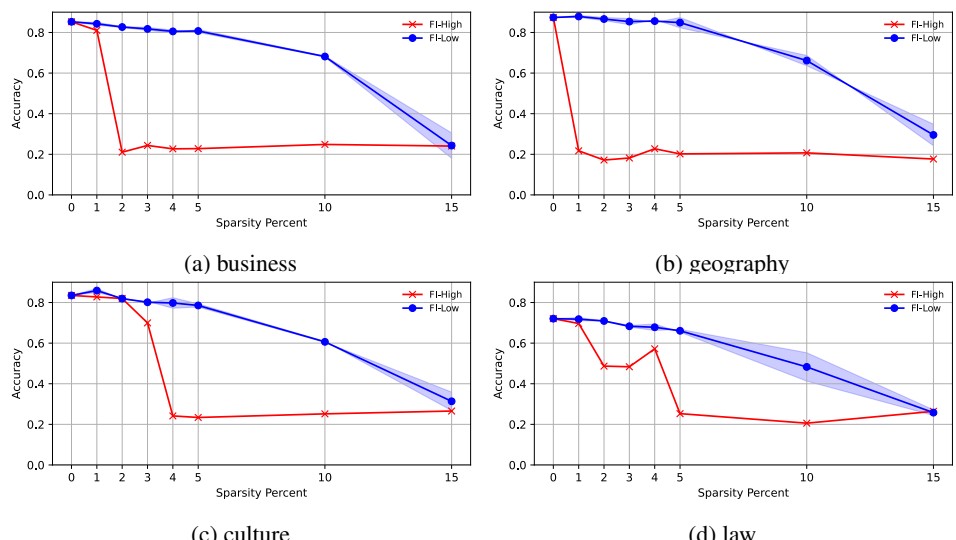

(a) business

(b) geography

(c) culture

(d) law

Figure 3: Comparing the accuracy in the MMLU dataset of Qwen2-7B when parameters have been sparsified at different rates.

effective optimization scheme based on FI. By designating a certain proportion of channels to remain at high precision, we seek to balance storage efficiency with computational efficacy.

Specifically, we conduct experiments with Qwen2-7B on the MMLU dataset, using the same cross-entropy objective function as in Section 3.2, but calculate FI values for each channel in the parameter matrix in line with standard practices in model quantization literature Xiao et al. (2023). We then protect a certain proportion of high-FI channels by maintaining them in FP16 precision while applying 1-bit or 4-bit quantization to the remaining parameters. Additionally, we compare the results of protecting low-FI channels. The results, as shown in Figure 5, demonstrate the effectiveness of this strategy under two different quantization precisions. Overall, it is evident that the protection strategy yields noticeable performance improvements across different protection ratios.

The selection of channels is crucial. Compared to randomly selecting low-FI channels, choosing high-FI channels consistently leads to better results. For instance, in the case of MMLU-Business with 1-bit quantization, randomly selecting low-FI channels fails to achieve performance improvements proportional to the protection ratio, whereas the high-FI strategy results in a 50% increase in accuracy. Even in relatively less challenging subjects such as MMLU-Business and Culture, protecting only 5% of the channels (rounded to 200) by keeping them in FP16 mitigates over 90% of the performance loss while requiring merely an additional 0.1 GB of memory.

**Application on Model Merging.** Model merging is another application scenario where parameter perturbations occur. It aims to acquire domain-specific knowledge by fusing models from different domains, thereby reducing the need for additional fine-tuning. However, a persistent challenge is that merging parameters can lead to the forgetting of original knowledge; as discussed in Section 3.2,

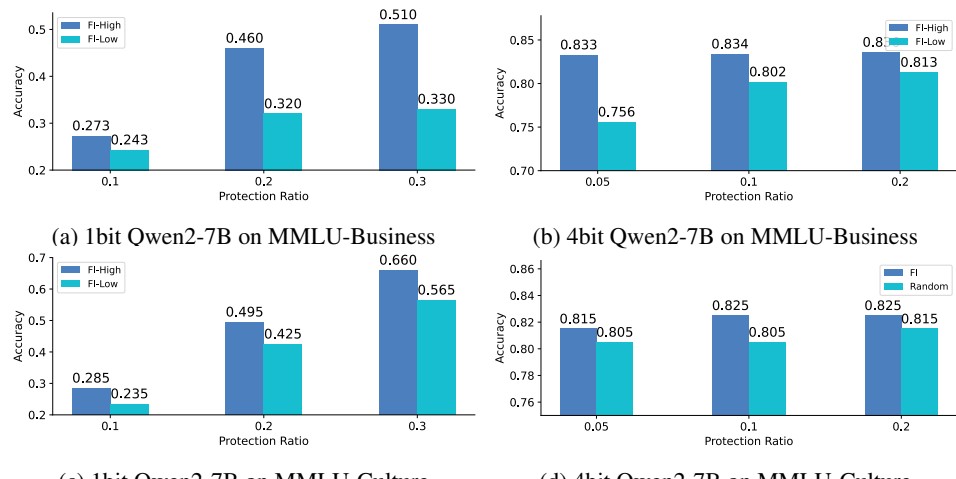

(a) 1bit Qwen2-7B on MMLU-Business

(b) 4bit Qwen2-7B on MMLU-Business

(c) 1bit Qwen2-7B on MMLU-Culture

(d) 4bit Qwen2-7B on MMLU-Culture

Figure 4: The accuracy obtained by preserving a certain proportion of Qwen2-7B o-proj channels at fp16 precision under 1-bit and 4-bit quantization methods.

parameter perturbations can impede a model's ability to retain previously learned information. Therefore, we utilize FI to identify parameters that may cause forgetting and exclude them from the merging process.

Specifically, we calculate the FI values for the parameters of the DeepSeek-Math (Shao et al., 2024) model on the MMLU-MATH dataset. Then, we merge the DeepSeek-Coder (Guo et al., 2024) model into it by arithmetic average, while simultaneously preserving the top 10% of high-FI parameters by excluding them from the merge. Finally, we evaluate the mathematical capabilities of the merged model on a series of mathematical benchmarks to verify whether knowledge forgetting is alleviated.

Table 2: Merge results

|  | Math-7b | Math&Code w/o Protect | Math&Code FI-High-Protect |
|---|---|---|---|
| GSM-8K | 0.811 | 0.739 | **0.752** |
| CMATH | 0.810 | 0.770 | **0.785** |
| MATH | 0.369 | 0.295 | **0.307** |
| MGSM | 0.766 | 0.653 | **0.672** |

As shown in Table 2, model merging results in a significant degradation in performance. However, by protecting parameters with high FI values, the model is able to recover some of the lost capabilities. The FI-Protect strategy leads to improved performance compared to random protection, with a 15-20% increase in performance across multiple benchmarks. This finding also indicates that FI can effectively determine parameter sensitivity based on domain knowledge rather than specific datasets, underscoring its practical value.

## 4 CONCLUSIONS

This paper unveils the presence of cracks in the armor of LLMs, which poses challenges for quantifying the stability of LLMs under perturbations. To address this issue, we introduce a novel framework for adversarial learning that evaluates the stability of each component within LLMs, considering the impact of small perturbations on the overall system. Through experiments under both internal and external perturbations, we demonstrate the effectiveness of our proposed method.

There are several promising avenues for future research. One direction is to develop methods that can accelerate the computation of the metric. Additionally, exploring the application of our FI to the training stage. By adapting and extending our FI metric, we can gain a deeper understanding of parameter stability during training, ultimately achieving more efficient parameter fine-tuning and pre-training.

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

# A    APPENDIX

## A.1    PROOF OF THEOREM 2.4

*Proof.* We apply Taylor expansion to $f(\omega(t))$ at the point $\omega(t)$:

$$f(\omega(t)) = f(\omega(0)) + \nabla_{f(\omega_0)}^T h_{\omega_0} t + \frac{1}{2} \left( h_{\omega_0}^T H_{f(\omega_0)} h_{\omega_0} + \nabla_{f(\omega_0)}^T d^2\omega(0)/dt^2 \right) t^2 + o\left(t^2\right), \quad (7)$$

where $\nabla_{f(\omega_0)} = \partial f(\omega)/\left.\partial\omega\right|_{\omega=\omega_0}$ and $H_{f(\omega_0)} = \partial^2 f(\omega)/\left.\partial\omega\partial\omega^T\right|_{\omega=\omega_0}$. From Equation 2, $S_C^2(\omega_t, \omega_0)$ can be approximated as $S_C^2(\omega_t, \omega_0) = t^2 h_{\omega_0}^T G_{\omega_0} h_{\omega_0} + o\left(t^2\right)$. Based on l'H^opital's rule, the influence measure FI from Equation2 can be rewritten as:

$$\mathbf{FI}_\omega\left(\omega_0\right) = \max_{h_\omega} \frac{h_\omega^T \nabla_{f(\omega_0)} \nabla_{f(\omega_0)}^T h_\omega}{h_\omega^T G_{\omega_0} h_\omega}.$$

We then reparameterize $\omega$ to $\tilde{\omega} = G_{\omega_0}^{-1/2}\omega$. According to Theorem 2.3, the influence measure **FI** remains invariant under this reparameterization

$$FI_\omega(\omega_0) = FI_{\tilde{\omega}}(\tilde{\omega}_0) = \arg\max_{h_{\tilde{\omega}}} \frac{h_{\tilde{\omega}}^\top G_{\omega_0}^{-1/2} \nabla_{f(\omega_0)} \nabla_{f(\omega_0)}^\top G_{\omega_0}^{-1/2} h_{\tilde{\omega}}}{h_{\tilde{\omega}}^\top h_{\tilde{\omega}}}.$$

The maximization problem is now in the form of a Rayleigh quotient, which attains its maximum when $h_{\tilde{\omega}}$ is proportional to $G_{\omega_0}^{-1/2} \nabla_{f(\omega_0)}$. Substituting back into the Rayleigh quotient, we find:

$$\mathbf{FI}_\omega(\omega_0) = \frac{\left(G_{\omega_0}^{-1/2}\nabla_{f(\omega_0)}\right)^T G_{\omega_0}^{-1/2}\nabla_{f(\omega_0)}\nabla_{f(\omega_0)}^T G_{\omega_0}^{-1/2}\left(G_{\omega_0}^{-1/2}\nabla_{f(\omega_0)}\right)}{\left(G_{\omega_0}^{-1/2}\nabla_{f(\omega_0)}\right)^T \left(G_{\omega_0}^{-1/2}\nabla_{f(\omega_0)}\right)}$$

$$= \frac{\nabla_{f(\omega_0)}^T G_{\omega_0}^{-1}\nabla_{f(\omega_0)}\nabla_{f(\omega_0)}^T G_{\omega_0}^{-1}\nabla_{f(\omega_0)}}{\nabla_{f(\omega_0)}^T G_{\omega_0}^{-1}\nabla_{f(\omega_0)}}$$

$$= \nabla_{f(\omega_0)}^T G_{\omega_0}^{-1}\nabla_{f(\omega_0)}.$$

This concludes the proof.                                                           □

## A.2    SUPPLEMENT OF EXPERIMENT A.2

In this section, we extend the experiment by applying the additional four different noise attacks on the image, including Poisson noise, random erasing, salt & pepper noise, and speckle noise. Each type of noise resulted in a decrease in the models' confidence. However, neither of them was able to induce incorrect answers. The detailed results are as follows:

- Baseline: bat star $p = 0.72$ and kelp $p = 0.28$

- Salt & Pepper noise: bat star $p = 0.64$ and kelp $p = 0.36$
- Speckle noise: bat star $p = 0.66$ and kelp $p = 0.34$
- Poisson noise: bat star $p = 0.67$ and kelp $p = 0.33$
- Random Erasing: bat star $p = 0.73$ and kelp $p = 0.27$

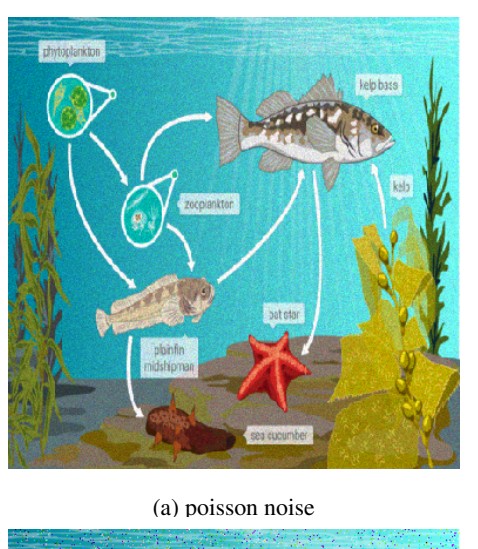

(a) poisson noise

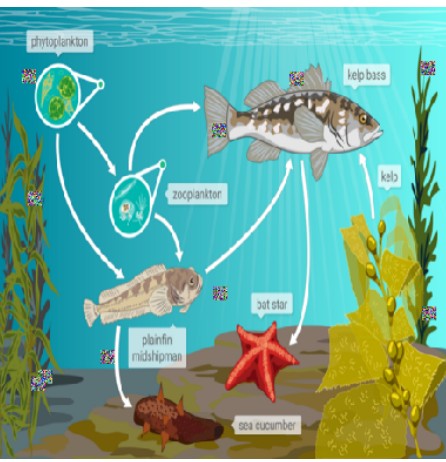

(b) random erasing

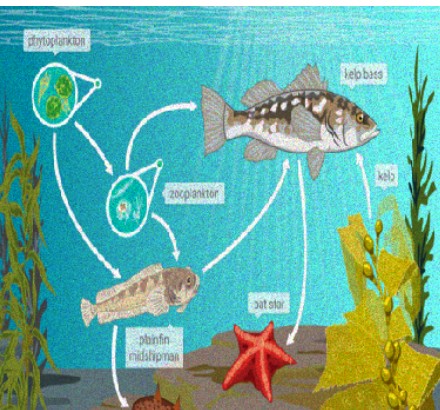

(c) salt & pepper noise

(d) speckle noise

Figure 5: Different noise attacks.

