# OpenReview forum: "Crack in the Armor: Universal Stability Measurement for Large Language Models"
_ICLR.cc/2025/Conference — ICLR 2025 Conference Withdrawn Submission_

### Official Review · Reviewer_f5Nc · 2024-10-18

**Soundness:** 2
**Presentation:** 2
**Contribution:** 2
**Rating:** 3
**Confidence:** 4

**Summary:**

The paper presents a new approach for understand VLM predictions, especially relating to their robustness to perturbation. Their metric (FI) essentially estimates the change in the model output with respect to input (or parameter) perturbations. The authors test their metric by identifying for a range of images the pixels that affect model predictions the most and altering them. Furthermore, the authors test their approach with respect to input parameters by identifying crucial parameters that should be left intact during quantization, and validating that performance deteriorates less when they're not changed.

**Strengths:**

- The proposed approach seems effective in identifying input regions/parameters that have a large effect on model output.
- The authors test their approach in a range of applications.
- I like the application of identifying sensitive parameters that should remain intact during quantization/sparsification.

**Weaknesses:**

- The paper does not compare against existing baselines.
- On the sensitivity to input pixels, how does this approach compare to Grad-CAM [1] and subsequent work? It is important to see a quantitative analysis.
- On the sensitivity to model parameters, it would be nice to see a comparison with existing approaches, e.g., [2], ...
- I feel there is too much going on in the paper: merging sensitivity to input images + parameters at the same time seems too much for a single project. I would suggest focusing on one and studying it in detail.
- Sensitivity of VLMs under different prompts is interesting but requires further analysis especially as to which changes in the prompts affect the influences, the semantic closeness of images and prompts, etc.


[1] Grad-CAM: Visual Explanations from Deep Networks via Gradient-based Localization, Selvaraju et al., 2016.
[2] Decomposing and Editing Predictions by Modeling Model Computation, Shah et al., 2024.

**Questions:**

See weaknesses.

- Can you provide more details about the compute cost of your approach?

---

### Official Review · Reviewer_z9Gk · 2024-10-31

**Soundness:** 3
**Presentation:** 3
**Contribution:** 3
**Rating:** 5
**Confidence:** 3

**Summary:**

This paper proposes a "First-order Local Influence" (FI) metric for quantifying LLM and VLM sensitivity to perturbations. By examining both internal (parameter) and external (input) perturbations, the FI metric aims to identify model weaknesses and improve robustness through selective parameter preservation. Experiments demonstrate FI’s potential in tasks like model quantization and merging.

**Strengths:**

+ FI offers a theoretically grounded approach to assessing parameter and input sensitivity, which can support robustness improvements across applications.

 + The paper provides experiments on various models, including applications of FI in quantization and model merging, demonstrating practical value.

**Weaknesses:**

- The paper is positioned primarily around LLMs and VLMs, but these stability concerns are more broadly applicable to general ML. A broader contextual framing would benefit the paper.

- The choice to protect high-FI parameters during distillation/model merging is questionable since some high-FI parameters might correspond to irrelevant or “nonsensical” inputs.

- Prior works on Fisher Information Matrix (FIM) in pruning and parameter sensitivity (e.g., Frantar & Alistarh, 2023; Yu et al., 2024) and Sharpness-Aware Minimization (SAM) (Foret et al., 2021) are not mentioned. These are relevant for contextualizing FI's robustness contributions.

**Questions:**

See Weaknesses

---

### Official Review · Reviewer_igMG · 2024-11-05

**Soundness:** 2
**Presentation:** 2
**Contribution:** 2
**Rating:** 3
**Confidence:** 3

**Summary:**

A white-box method for identifying the most important/salient regions of an input for making a prediction is presented. The paper          describes the method using the formalism of differential geometry and apply it to VLMs and LLMs. For VLMs, they apply their method to      sensitivity analysis For LLMs, they apply their method to model quantization and model merging.

**Strengths:**

The paper presents interesting and novel applications of saliency maps to model quantization and model merging.

The paper strongly motivates the usage of saliency maps.

**Weaknesses:**

I currently suggest to reject this paper on the basis that I don't understand the novelty of the method and due to the lack of comparisons with the baselines. I am open to changing my mind,  but I strongly encourage the authors to focus on those specific points in their response and be very clear how their method differs from existing works.

Novelty of the method: Not clear how the proposed method is different from other analysis methods such as Saliancy maps [1] and            adversarial perturbations. A throughout analysis of the related methods should be presented and compared against in the paper.

Comparison with the baselines: Several compelling applications of the method are proposed, but no comparison to existing baseline methods tacking these applications. The authors should consider comparing the presented method against relevant baselines on standard benchmarks so that the reader can assess the usefulness of the method.

Clarity of the paper: I found the paper to be hard to follow. The paper introduces unnecessarily abstracts notions to describe the method. I   don't understand why such abstraction is needed to describe the idea presented in the paper. Moreover, a lot of terms a unnecessarily defined. For example,  $l(\omega|y,x,theta)$ could be written as $\log P(y|x,\theta,\omega)$ and $f(\omega)$ as $-P(y_\text{pred}|x,\theta,\omega$ and it would make the reading clearer. Some terms like $h_j$ are not clearly defined.

[1] https://arxiv.org/abs/1312.6034

**Questions:**

* How does the method presented in the paper differs from existing works?
* Figure 3, Table 1, Figure 4: A baseline where the parameters are randomly selected is needed. What are the performances of such a        baseline?
* Table 1, Figure 3: How does the method compares to other pruning methods such as the one presented in this survey [2]
* Table 2: How does the method compares to other model merging method such as the one presented in this survey [3]


[2] https://arxiv.org/pdf/2308.06767
[3] https://arxiv.org/pdf/2408.07666

---

### Note · Authors · 2024-11-26

I have read and agree with the venue's withdrawal policy on behalf of myself and my co-authors.